# *In Vivo* Evaluation of Permeable and Impermeable Membranes for Guided Bone Regeneration

**DOI:** 10.3390/membranes12070711

**Published:** 2022-07-15

**Authors:** Suelen Cristina Sartoretto, Natalia de Freitas Gens, Rodrigo Figueiredo de Brito Resende, Adriana Terezinha Neves Novellino Alves, Rafael Cury Cecato, Marcelo José Uzeda, Jose Mauro Granjeiro, Monica Diuana Calasans-Maia, Jose Albuquerque Calasans-Maia

**Affiliations:** 1Oral Surgery Department, Dentistry School, Fluminense Federal University, Niteroi 24020-140, Rio de Janeiro, Brazil; susartoretto@hotmail.com (S.C.S.); rodrigoodonto@yahoo.com.br (R.F.d.B.R.); mjuzeda@gmail.com (M.J.U.); 2Laboratory for Clinical Research in Dentistry, Dentistry School, Fluminense Federal University, Niteroi 24020-140, Rio de Janeiro, Brazil; aterezinhanovellino@gmail.com (A.T.N.N.A.); jmgranjeiro@gmail.com (J.M.G.); monicacalasansmaia@gmail.com (M.D.C.-M.); 3Graduate Program, Dentistry School, Fluminense Federal University, Niteroi 24020-140, Rio de Janeiro, Brazil; ngens@id.uff.br; 4Oral Surgery Department, Dentistry School, Iguaçu University, Nova Iguaçu 26275-580, Rio de Janeiro, Brazil; 5Oral Diagnosis Department, Dentistry School, Fluminense Federal University, Niteroi 24020-140, Rio de Janeiro, Brazil; 6Implant Dentistry Center for Education and Research on Dental Implants (CEPID), Department of Dentistry, Federal University of Santa Catarina (UFSC), Florianópolis 88000-000, Santa Catarina, Brazil; rafael.cury.cecato@posgrad.ufsc.br; 7National Institute of Metrology, Quality and Technology (INMETRO), Duque de Caxias 25000-000, Rio de Janeiro, Brazil; 8Orthodontic Department, Dentistry School, Fluminense Federal University, Niteroi 24020-140, Rio de Janeiro, Brazil

**Keywords:** biocompatibility, membranes, subcutaneous, mice, PLGA, PTMC

## Abstract

Background: The degree of biodegradation and the inflammatory response of membranes employed for guided bone regeneration directly impact the outcome of this technique. This study aimed to evaluate four different experimental versions of Poly (L-lactate-co-Trimethylene Carbonate) (PTMC) + Poly (L-lactate-co-glycolate) (PLGA) membranes, implanted in mouse subcutaneous tissue, compared to a commercially available membrane and a Sham group. Methods: Sixty Balb-C mice were randomly divided into six experimental groups and subdivided into 1, 3, 6 and 12 weeks (n = 5 groups/period). The membranes (1 cm^2^) were implanted in the subcutaneous back tissue of the animals. The samples were obtained for descriptive and semiquantitative histological evaluation (ISO 10993-6). Results: G1 and G4 allowed tissue adhesion and the permeation of inflammatory cells over time and showed greater phagocytic activity and permeability. G2 and G3 detached from the tissue in one and three weeks; however, in the more extended periods, they presented a rectilinear and homogeneous aspect and were not absorbed. G2 had a major inflammatory reaction. G5 was almost completely absorbed after 12 weeks. Conclusions: The membranes are considered biocompatible. G5 showed a higher degree of biosorption, followed by G1 and G4. G2 and G3 are considered non-absorbable in the studied periods.

## 1. Introduction

The concept of guided bone regeneration (GBR) treatment was developed based on the principle of guided tissue regeneration (GTR), which was introduced in the mid-1980s. In GTR, regeneration of a particular tissue type is achieved when cells with the ability to regenerate a specific lost-tissue type are allowed to fill the defect during healing [1,2,3,4]. The rationale of biological GBR is the mechanical exclusion of unwanted soft tissue growth in the bone defect, allowing osteogenic cell populations derived from native bone to inhabit the bone defect [3,5].

The therapeutic protocol of GBR involves the surgical implantation of an occlusive membrane facing the bone surface to physically seal the bone defect [3]. In addition, the membrane creates and maintains an isolated space permissive for the recruitment and proliferation of these cells, differentiation along the osteoblast lineage, and the expression of osteogenic activity [6,7]. Various resorbable and non-resorbable membranes have been used in experimental studies and clinical trials in the context of GBR. The desirable characteristics of membrane barriers used for GBR therapy include biocompatibility, cell occlusion properties, integration by host tissues, clinical handling, and space maintenance ability [7].

The main class of non-absorbable membranes is made of polytetrafluoroethylene (e-PTFE), a non-degradable material. e-PTFE is chemically stable and biologically inert and has a porous structure and flexible shape. It resists microbiological and enzymatic degradation, does not provoke an immune response [8], and has been frequently used for periodontal and bone regeneration. These non-resorbable membrane barriers do not solubilize when in contact with the living organism, requiring a second surgical intervention to be removed.

Several biodegradable materials have been tested successfully in bone and periodontal regeneration. Most GBR membranes are made of animal-derived collagen (i.e., porcine or bovine) [9,10]. The biodegradation of collagen membranes depends on the tissue of origin and chemical cross-linking. As a result, these membranes do not maintain their mechanical integrity over long periods and may often collapse over bone defects due to early resorption [11].

Other biodegradable membranes that are commercially available are polyurethane, polyglactin 910, polylactic acid, polyglycolic acid, polyorthoester, and different copolymers of polylactic and polyglycolic acid [12,13,14]. When deployed in aqueous environments, such as a biological system, biodegradable polymers go through four stages: hydration, loss of strength, loss of mass integrity, and solubilization through phagocytosis. The time duration of each step and the rate of degradation will depend on the nature of the polymer, the pH, the temperature, the degree of crystallization of the polymer, and the volume of the membrane [15,16]. It can be concluded that the function of the barrier duration is not strictly controlled, and that the resorption process can interfere with the bone defect regeneration process [15].

In this study, we evaluate an alternative material for GBR using a new synthetic and biodegradable membrane made of PLGA (poly-D, L-lactic/glycolic acid) [9,17] and Poly (L-lactate-co-Trimethylene Carbonate) (PTMC). PLGA is a well-known biodegradable polymer and is widely used in other medical applications (e.g., resorbable sutures and surgical meshes) [9]. PTMC is a biomedical polymer that is considered low-toxicity, biocompatible, and biodegradable, has good flexibility and good surface erosion properties, and does not produce strong acid compounds after degradation [18,19,20].

The association of these polymers (PLGA + PTMC) was performed to obtain a membrane with controlled degradation time, compatible with the need for tissue isolation required in GBR techniques, in addition to a balanced inflammatory reaction. The PTMC composition seems to present a high degradation time [21,22], while PLGA seems to present a lower degradation time [23]. Additionally, this composition creates a flexible and resistant membrane, even after hydration and without shape memory, allowing adequate adaptation to the surgical bed.

The aim of this study was to evaluate the biocompatibility and biosorption of four different types of PLGA + PTMC membranes, implanted in mouse subcutaneous tissue, when compared to a commercially available membrane and a Sham group (no implantation), according to ISO 10993-6 scores (Figure 1).

## 2. Materials and Methods

### 2.1. Ethical Considerations

The research protocol for this study was approved by the Ethics Committee on Animal Use of the Universidade Federal Fluminense on 14 November 2019 (CEUA/UFF No. 2821220919). This research was conducted following the Brazilian Guidelines for the Care and Use of Animals for Scientific and Teaching Purposes-DBCA and the CONCEA Guidelines (Conselho Nacional de Controle e Experimentação Animal) for the Practice of Euthanasia.

Additionally, the investigation was carried out in observance of the guidelines of the 3Rs Program (Reduction, Refinement, and Replacement), whose goal is to reduce the number of animals used during experimentation and to minimize their pain and discomfort (National Centre for the Replacement, Refinement & Reduction of Animals in Research—NC3Rs, 2010) [24].

The results were documented according to the ARRIVE (Animal Research: Reporting of In Vivo Experiments) [25] and Planning Research and PREPARE (Experimental Procedures on Animals: Recommendations for Excellence) [26] guidelines concerning relevant items.

### 2.2. Materials

In this study, permeable and impermeable PTMC + PLGA membranes were compared. The permeable polymeric membranes of randomly deposited fibers formed a three-dimensional mesh. In the manufacturing process, the polymer is dissolved in an organic solvent and subjected to a high (negative) electric field. When the applied electric field overcomes the surface tension and viscous forces of the polymer solution, a charged jet of the solution is ejected toward the opposite pole (collector), also electronically charged but with the opposite charge (positive), thus forming the fibers. In the path between the polymer solution and the fiber collection surface, the jet presents regions of instability, evaporating the solvent and depositing solid fibers on the collector (electrospinning). Then, the resulting membrane is removed from the collector and cut according to the need and indication for use. The impermeable polymeric barriers were manufactured by preparing a solution of the polymer in an organic solvent. After total homogenization of the solution, it is placed in polyacetal molds with the internal thickness required for the barrier. The molds are vacuumed for 3 h for complete evaporation of the solvent and drying of the barriers. Then, the drying process is completed, and the barriers are removed from the molds and ready for cutting, according to the need and indication for use (manufacturing by random fiber deposition). The permeable and impermeable PTMC + PLGA membranes were divided into the following experimental groups: Group 1: 100 µm thickness permeable (batch 150819-2); Group 2: 100 µm thickness impermeable (batch 100919); Group 3: 70 µm thickness impermeable (batch 200919); Group 4: 250 µm thickness permeable (batch 150819-1); Group 5: commercially available Gore Bio-A^®^ (batch 18679527); and Group 6: Sham (without implantation). The Group 5 sample with already commercially available material was selected for its equivalent composition, which is Polyglycolate (PLG) and PTMC (Figure 2).

### 2.3. Physico-Chemical Characterization of the Membranes

The analyses of the morphologies of samples were performed by employing scanning electron microscopes with field emission sources (SEM) (model JSM-6701F-JEOL LTD-Japan) at the State University of Santa Catarina (UDESC). The samples were prepared for analysis by attaching them to the equipment using conductive carbon tape. They were then sprayed with gold on the surface to enable the conduction of electrons for the formation of images. Each sample was identified according to its specific group. After placing the samples in the equipment, scanning was performed at an acceleration voltage of 15 kV. Images were captured at 150× and 1000× magnification to visualize the integrity of the polymeric fibers.

Fourier transform infrared spectroscopy (FTIR) was performed at FGM Company (FGM Dental Group) (Spectrum 100-Perkin Elmer-USA) to show the types of chemical bonds present in the analyzed substances. Using FTIR makes it possible to characterize the product and track whether there have been changes in the chemical bonds, which, in the case of polymeric materials, indicates the degradation of the polymer. One membrane per group (1 × 1 cm) was used for FTIR evaluation. The membranes were arranged on the infrared crystal and compressed at 100 N to overcome the radiation and to obtain data for the formation of the FTIR bands. No KBr was used in the preparation of the samples.

### 2.4. Animal Characterization

In this study, 60 male and female Balb-C mice were used. They were approximately 50 days old, weighing 20 to 30 g, and were provided by the Laboratory of Animal Center of the Universidade Federal Fluminense (NAL-UFF). Before and after the experimental period, the animals were kept in isolators with a maximum of five animals each and were fed with pelleted feed (Nuvilab^®^, Curitiba, Brazil) and water ad libitum. The animals were divided through a random draw (using an opaque envelope containing the group name) by the principal investigator into six experimental groups, i.e., Group 1: 100 µm thickness permeable; Group 2: 100 µm thickness impermeable; Group 3: 70 µm thickness impermeable; Group 4: 250 µm thickness permeable; Group 5: commercially available Gore Bio-A^®^; and Group 6: Sham (without implantation), and periods of evaluation (1, 3, 6, and 12 weeks), with five animals in each group/period.

With the ethical objective of reducing the number of animals and according to the 3Rs Program guidelines [24], two incisions were made in the back of each animal (right and left sides). This study was performed according to ISO 10993-6/2016 [27].

A significance level of 5% and a power test of 80% were used to calculate the sample size used in this study (Sealed Envelope. Available online: https://www.stat.ubc.ca/~rollin/stats/ssize/n2.html Accessed on 15 May 2022). The results suggested five animals in each group.

### 2.5. Surgical Procedures

After fasting for 24 h, all animals were submitted to intraperitoneal general anesthesia following the protocol of the Laboratory of Animal Experimentation (LAE) of UFF with the injection of 0.6 mL of an anesthetic solution prepared with 1.0 mL of 10% ketamine (Dopalen^®^-100 mg/mL Ceva, Paulina, SP, Brazil), 0.5 mL of 2% xylazine (Anasedan^®^-20 mg/mL Ceva, Paulina, SP, Brazil), and 8.5 mL of sterile saline solution (KabiPac^®^ Fresenius Kabi Brasil Ltd., Barueri, SP, Brazil). Approximately three minutes later, trichotomy and degerming were performed with chlorhexidine degerming solution and 2% alcoholic chlorhexidine (Rioquímica, São José do Rio Preto, SP, Brasil), followed by the apposition of previously sterilized surgical drapes, for the delimitation and isolation of the surgical site. A rectilinear incision was made using a no. 3 scalpel cable (Bard Parker^®^, Aspen Surgical, Caledonia, MI, USA) with blade no. 15C (Becton-Dickinson^®^, Juiz de Fora, MG, Brazil) on each side of the dorsal region of the animal with about 10 mm of extension in the skin region, followed by displacement of the skin of the muscle fascia with the aid of blunt-point scissors (Golgran^®^, São Caetano do Sul, SP, Brazil), exposing the subcutaneous tissue for insertion of the membrane (1 cm) in the subcutaneous region, followed by suture with 5.0 nylon thread (Ethicon^®^, Johnson and Johnson, São Paulo, SP, Brazil) and antisepsis with gauze and alcoholic chlorhexidine solution (Rioquímica, São José do Rio Preto, SP, Brasil). In the postoperative period, the animals were kept at the LAE, divided into boxes by their experimental groups, where they received free food and water. Meloxicam 5 mg/kg was administered subcutaneously every 24 h (Medley, Curitiba, PR, Brazil) on the day of surgery and on the two subsequent days.

### 2.6. Sample Obtention

After the experimental periods of 1, 3, 6, and 12 weeks, the animals in each experimental group received a lethal dose of general anesthetic to collect samples and surrounding tissues with a 5 mm safety margin. After collection, the pieces were kept in 4% formalin solution (phosphate buffer, pH 7.4) for 48 h for fixation, and then the samples were sent for histological processing. All samples obtained were fixed, decalcified, dehydrated, clarified, and embedded in paraffin to obtain 5 μm thick sections. The slides were stained with Hematoxylin and Eosin for descriptive and semiquantitative histological evaluation.

### 2.7. Microscopic Evaluation

The slides were obtained from the paraffin-embedded blocks and stained with HE. All microscopic analyses were performed by a single experienced pathologist who was blinded through coded slides. The slides were observed under a brightfield light microscope (OLYMPUS BX43, Tokyo, Japan). These images were captured by a high-resolution digital camera (OLYMPUS SC100, Tokyo, Japan) at the Laboratory of Applied Biotechnology (LABAHisto) at UFF.

#### 2.7.1. Histological Description

The descriptive analysis of the tissue response to the membranes was evaluated according to the presence of the membrane, its biosorption/degradation pattern, the membrane–tissue interface, and the presence, amount, and type of inflammatory infiltrate. A magnification of 40× was applied to obtain more comprehensive visualization of the area of interest, and 200× and 400× magnification lenses were to obtain cellular and tissue details.

#### 2.7.2. Semiquantitative Evaluation of Inflammatory Infiltrate

Semiquantitative histological analysis was conducted on each slide, from which 10 fields were scanned according to the area of interest, without any overlap, and captured using high-resolution software (CELLSENS^®^1.9 DigitalImage, Olympus, Tokyo, Japan) with an objective of 400×.

The biological response parameters at the tissue membrane interface were evaluated according to Annex E (Tables E1 and E2) of ISO 10993-6:2016 [27] and scored as follows: (1)The quantity and distribution of inflammatory cells present at the tissue–material interface. The neutrophils, lymphocytes, plasma cells, and macrophages were classified as: absent (score zero), rare or 1–5 per high-powered (400×) field (phf) (score 1), 5–10 phf (score 2), heavy infiltrate (score 3), and packed (score 4). The multinucleated cells were classified as: absent (score zero), 1–2 per high-powered (400×) field (phf) (score 1), 3–5 phf (score 2), heavy infiltrate (score 3), and sheets (score 4).(2)The inflammatory response parameters (neovascularization, the degree of fibrosis, and fatty infiltrate.(3)The presence of necrosis.

For each histological characteristic evaluated (such as the presence of polymorphonuclear cells, giant cells, plasma cells, and/or degradation of material), the semiquantitative scoring system was described in the evaluation report. In addition to the scoring of the reaction components, the extent of the whole reaction was also evaluated and compared with the Sham group.

## 3. Results

### 3.1. Scanning Electron Microscopy (SEM)

The samples were analyzed by scanning electron microscopy (SEM) at 150× and 1.000×. Group 1 presented a homogeneous surface with randomly arranged fibers with irregular diameters forming irregular pores (150×); at 1000× magnification, we observed fibers with varying diameters forming irregular, rough, and interconnected pores. In Group 2, a homogeneous and flat surface without characterizations was observed. The higher magnification revealed the presence of marks, probably caused by bubbles (black spots) during the manufacturing process. Group 3 presented similar patterns of surface and fibers to Group 2. Group 4 was equivalent to Group 1. Group 5 presented a homogeneous surface with randomly arranged fibers of regular diameter forming regular pores. At 1000×, fibers with a regular diameter of about 30 m and regular interconnected pores were observed (Figure 3).

### 3.2. Fourier Transform Infrared Spectrometer (FTIR)

The FTIR spectrum showed similar vibrational modes of typical PTMC and PLGA compounds to the samples of Groups 1 to 4. In addition, the sample of Group 5 showed similar vibrational modes to a typical PLG compound (Figure 4).

### 3.3. In Vivo Response to Membranes and Macroscopy Results

All animals recovered well after the surgical procedures. During the healing periods (1, 3, 6, and 12 weeks), no animals died, and no complications that could compromise the results were detected.

Figure 5 presents the macroscopic aspect of tissue surrounding the implanted membranes after 12 weeks. The membranes in Groups 1, 2, 3, and 4 were clearly visible when the subcutaneous tissue was exposed. In Group 5, during sample collection, only granulation tissue surrounding the membrane was observed. Clinically, the membrane appeared to have been resorbed. In the Sham group, tissue without alterations was observed.

### 3.4. Descriptive Histological Analysis

The microscopic aspect of membranes and the biological tissue response post-implantation are shown in Figure 6 (1 week after implantation), 7 (3 weeks), 8 (6 weeks), and 9 (12 weeks).

#### 3.4.1. One Week Post-Implantation

Group 1: In G1, we observed the presence of the membrane with a fibrillar aspect and permeated with mesenchymal cells (fibroblasts), mononuclear inflammatory cells and scarce polymorphonuclear cells, and abundant multinucleated giant cells (MNGCs), both in the periphery and permeating the membrane. Some MNGCs had material in their interior. Surrounding the membrane was a narrow band of granulation reaction of predominantly mononuclear inflammatory infiltrate (Figure 6A,B).Group 2: In all animals in this group, it was not possible to visualize the membrane: a dense band of connective tissue containing an intense mixed inflammatory infiltrate was observed (Figure 6C,D). The membrane appeared not to have been incorporated into the underlying tissues and detached during histological processing. It was possible to observe the contour or virtual space that the membrane occupied.Group 3: In all animals in this group, no membrane was identified, similarly to Group 1. The membrane appeared not to have been incorporated into the underlying tissues and detached during histological processing. Additionally, in the contour, it was possible to observe connective tissue containing moderate mixed inflammatory infiltrate (Figure 6E,F).Group 4: In this group, we observed the presence of a thick membrane (M) with a fibrillar aspect, covered by a narrow band of connective tissue with a mild to moderate granulation reaction. Multinucleated giant cells were observed in its periphery, sometimes containing material inside. Permeating the membrane, there was the presence of fusiform and starred mesenchymal cells (Figure 6G,H).Group 5: The presence of a thick membrane with a spherical and tubular aspect, peripherally interspersed with delicate connective tissue bundles, was observed. We noticed cell adherence on the spherical structures of the membranes and a few multinucleated giant cells. In the membrane, we observed connective tissue with a mild inflammatory infiltrate (Figure 6I,J).Group 6: The dermis was lined by orthokeratinized stratified squamous epithelium exhibiting areas of extensive keratinization and scar hyperplasia; underlying fibrous connective tissue was observed with a focal area of intense mixed inflammation and collagen and muscle fibrous connective tissue with focal areas of intense mixed inflammation and organized collagen and muscle fibers (Figure 6K,L).

#### 3.4.2. Three Weeks Post-Implantation

Group 1: Membranes with a fibrillar appearance and permeating mesenchymal cells (fibroblasts), mononuclear inflammatory cells, and scarce polymorphs were observed, along with abundant multinucleated giant cells (MNGCs), both at the periphery and permeating the membrane, containing content. Some MNGCs displayed material inside. Integrated in the membrane was a delicate connective tissue with a few inflammatory cells (Figure 7A,B).Group 2: In G2, a rectilinear, homogeneous, and matte membrane appearing “loose” in almost the whole sample, without adherence to the connective tissue, was observed. In its periphery, multinucleated giant cells and mononuclear cells were trying to break through the membrane. A dense band of the granulation reaction in the connective tissue was subjacent (Figure 7C,D).Group 3: In this group, the membrane was rectilinear, homogeneous, matte, and “loose,” without adherence to the connective tissue. Multinucleated giant cells and mononuclear cells were present in its periphery, similarly to Group 2. In proximity to the membrane, there was connective tissue with a moderate mixed inflammatory infiltrate (Figure 7E,F).Group 4: We observed the presence of a thick fibrillar membrane permeated by sparse mesenchymal cells with a fusiform and stellate appearance. In the periphery, there was a narrow band of connective tissue with the granulation reaction, composed of a small amount of mononuclear inflammatory infiltrate with a predominance of macrophages. Multinucleated giant cells in its periphery were observed (Figure 7G,H).Group 5: A thick membrane with a spherical and tubular aspect interspersed with delicate connective tissue bundles was observed; cellular adherence on its surface was observed in some “spheres” of the membrane with few CGMNs. We observed connective tissue with mild inflammatory infiltrate integrated into the membrane (Figure 7I,J).Group 6: In the incision area, a reorganization of fibrous and muscular tissues interspersed with a moderate mononuclear inflammatory infiltrate was noted (Figure 7K,L).

#### 3.4.3. Six Weeks Post-Implantation

Group 1: In this group, it was observed that, six weeks after implantation, the membrane was interspersed with loosely arranged connective tissue and multinucleated giant cells with contents inside. The membrane was present in all animals (Figure 8A,B). The membrane, despite being permeated by a cell population and connective tissue, maintained its scaffold.Group 2: The presence of a rectilinear and homogeneous membrane with a microscopic matte appearance and without adherence to the adjacent connective tissue was observed. In the connective tissue surrounding the membrane, a granulation reaction with intense mononuclear infiltration also was observed, in addition to the presence of multinucleated giant cells in the periphery of the membrane (Figure 8C,D).Group 3: The membrane of G3 had a matte, rectilinear, and homogeneous aspect. Microscopically, it looked partially loose, without adherence to the adjacent connective tissue, limited by mild multinucleated giant cells and mononuclear cells. In the adjacent membrane, connective tissue with inflammatory cells was present (Figure 8E,F).Group 4: A thick membrane with a fibrillar aspect containing in its periphery a narrow range of the granulation reaction and multinucleated giant cells with content inside was observed. It is possible that the membrane was permeated by mesenchymal cells with a fusiform and stellate aspect (Figure 8G,H).Group 5: In this group, we observed a membrane composed of multiple structures with a clear basophilic spheroidal appearance bounded by multinucleated giant cells and macrophages. Fibrocellular connective tissue permeated the spheres (Figure 8I,J).Group 6: In this group, organized connective tissue with sparse inflammatory cells was observed. Adipose, muscular, and glandular tissues were present in the region (Figure 8K,L).

#### 3.4.4. Twelve Weeks Post-Implantation

Group 1: In this group, the membrane was surrounded by loosely arranged connective tissue with scant inflammatory cells and giant cells in the periphery. At the highest enlargement of the membrane, it was observed that multinucleated giant cells in the membrane presented contents inside, and mesenchymal cells were present permeating the membrane (Figure 9A,B). As in the case at six weeks, the membrane, although permeated by a cell population and connective tissue, had maintained its framework.Group 2: The membrane appeared rectilinear, intact, homogeneous, and, with a matte aspect, “loose”, without adherence to the connective tissue. In its periphery was observed the presence of multinucleated giant cells and mononuclear cells inside the membrane. A band of the granulation reaction was found underneath the membrane (Figure 9C,D).Group 3: The membrane was rectilinear, intact, homogeneous, matte, and “loose”, with no adherence to the connective tissue. In its periphery, the presence of multinucleated giant cells and a few mononuclear cells were observed. In proximity to the membrane, connective tissue with a few inflammatory cells was noted (Figure 9E,F).Group 4: In this group, the membrane had a fibrillar aspect and, contained in the periphery, a sparse mononuclear inflammatory infiltrate, and a large number of multinucleated giant cells were observed. It was possible to observe, at the highest magnification, the membrane permeated by mesenchymal cells with a fusiform and striated aspect and, in the periphery, multinucleated giant cells with content inside (Figure 9G,H).Group 5: The presence of a few membrane fragments interspersed with a band of connective tissue exhibiting granulation reaction was observed. From the details at the higher magnification, it was possible to observe membrane residues interspersed with the granulation reaction and multinucleated giant cells (Figure 9I,J).Group 6: In this group, well-organized tissue with scarce inflammatory cells and a normal healing appearance was observed for the 12-week period (Figure 9K,L).

### 3.5. Semiquantitative Histological Analysis of Local Biological Effect of Implanted Membranes: ISO 10993-6: 2016/Part 6/Annex E

According to ISO 10993-6, scores were established to classify the inflammatory cell response, including polymorphonuclear cells, lymphocytes, plasma cells, macrophages, and giant cells (shown in Figure 10A–E). Additionally, the overall tissue response according to inflammatory parameters, such as neovascularization, was evaluated (Figure 10F). The fibrosis, fatty infiltrate, and the degree of degeneration (necrosis) were also assessed; however, none of the studied groups showed signs of these events.

#### 3.5.1. PMN Cells

One week after implantation, G2 and G3 recruited more PMN cells than Group 4 (*p* = 0.004). After three weeks, Group 3 showed a tendency towards a greater population of PMNs than the other groups and was different from the Sham group (0.001). After six weeks, Groups 1 (*p* = 0.025) and 2 (*p* = 0.007) showed greater cell recruitment than the G5 and Sham groups. The G1, G5, and Sham groups showed a time-dependent reduction in PMN cell volume at 6 and 12 weeks (*p* < 0.04) (Figure 10A).

It is important to point out that the maximum score (4—packed) was not reached in any experimental group/period. In addition, in one week, while Groups 2 and 3 reached a score of 3, they did not show differences compared to the Sham group. In other words, against the different membranes, the tissue responded mildly, without acute inflammation or infection.

#### 3.5.2. Lymphocytes

In the first week after implantation, G2 and G3 had a greater volume of lymphocytes than the Sham group (*p* < 0.007). In addition, Group 2 was higher than Groups 1, 4, and 5 (*p* = 0.031). The greater recruitment of lymphocytes in Groups 2 and 3 when compared to Sham remained for a period of three weeks (*p* = 0.004). After six weeks, the G2 difference remained present compared to Sham (*p* = 0.002). The difference was also observed between G5 and the Sham group (*p* = 0.04). After 12 weeks, Group 2 continued to present a greater volume of lymphocyte cells compared to Sham (*p* = 0.006). In addition, G5 showed higher rates than Sham (*p* = 0.04).

Like neutrophils, the lymphocytes did not show a maximum score in any group/experimental period. High levels of mononuclear cell recruitment were observed in G2, which maintained a heavy infiltrate score during all experimental periods (*p* < 0.01). This reaction can be explained by the greater cell signaling in membranes with difficult incorporation/phagocytosis, consequently causing a more prominent and persistent granulation reaction (Figure 10B).

#### 3.5.3. Plasma Cells

One week after implantation, G5 had a greater volume of plasma cells than G2 and G3 (*p* = 0.02). Even so, the score was considered low (score 1). G1 had a maintained cell population in all experimental groups (score = 1) and, after 6 (*p* = 0.02) and 12 weeks (0.0001), showed differences from most of the other groups, probably due to the resorption activity that was observed in all experimental periods (Figure 10C).

#### 3.5.4. Macrophages

In the first week of implantation, there was no difference among the experimental groups. All membranes showed a macrophage population like the Sham group. After three weeks, except for Group 5, all groups showed higher macrophage activity than Sham (*p* = 0.002) (Figure 10D). After 6 weeks, there was a jump in macrophage volume exclusively in G5 compared to Groups 1 (*p* = 0.01) and 3 (*p* = 0.04) and especially compared to Sham (*p* < 0.0001), indicating their high resorption rates at 6 weeks and in the following period (12 weeks). In the last experimental period, Groups 2 and 4 showed greater phagocytic activity than the Sham group (≤0.003).

#### 3.5.5. Giant Cells

In all experimental periods, G1 (*p* < 0.003) and G4 (*p* < 0.001) had a higher number of giant cells compared to the Sham group (score of 3 = heavy infiltrate, in all periods). This result indicates a greater capacity of phagocytosis/resorption of membranes in G1 and G4, corroborating the descriptive histological results, which show higher rates of cellular invasion and reabsorption in the membranes of Groups 1 and 4 (Figure 10E).

Furthermore, in the first experimental period, Groups 1 and 4 presented a greater volume of giant cells than Group 2. In addition, G2 and G3 had a score of 0 and 1, respectively, within one week, probably due to the delay in incorporating membranes into the tissue seen in histology. In subsequent periods, the giant cell population gradually increased in these groups (*p* < 0.01; 12 and 6 weeks versus 1 week).

An increase in the giant cell count in Group 5 after six weeks was also observed, showing a difference from the Sham group (*p* = 0.001). This event demonstrates the resorptive activity in G5, corroborating the findings of macrophage values and histological description.

#### 3.5.6. Neovascularization

The Sham group had a score = 1 (minimal capillary proliferation, 1 to 3 focal buds) in all experimental periods (Figure 10F). In the one-week period, Groups 2 and 3 had a higher NV volume than the Sham group (*p* = 0.006). After three weeks, this event occurred in G1 and G2 (*p* = 0.01). After six weeks, only Group 2 had a score = 3, which was statistically different from the Sham group (0.009). Finally, at 12 weeks, Group 2 and Group 4 were higher than Sham (*p* < 0.03).

## 4. Discussion

Guided bone regeneration (GBR) is a routine practice in implant dentistry. It consists of using a membrane as a barrier that may or may not be associated with substitute biomaterials over a defect before primary closure to control tissue growth. This technique has been used for decades and aims to allow the cellular neoformation of the desired tissue, preventing the development of other undesirable cell types [10,17,28,29]. For the application of the concept of GBR, the membrane must be positioned in place and left for a period to ensure space for bone repair and avoid the invasion of connective tissue into the area [30,31].

In this context, the key element of GBR is the membrane and its ability to act as a barrier. To evaluate the biological performance of PLGA + PTMC membranes with an indication for GBR, this study used the subcutaneous tissue of rats as an experimental site. This model is widely used in material biocompatibility assessment studies [17,32,33], since it is considered easy to handle, low-cost, and safe. In addition, this surgical site is standardized by ISO 10993-6/2016 [10,33,34] to assess the biocompatibility of biomaterials by analyzing the recruitment of inflammatory cells and also allows the understanding of how the tissue responds to the presence of membranes, as well as the chronological events of biodegradation [17,27]. Considering the concepts of translational research, the objective of this study is not to extrapolate the results for human use but to comprehend the behavior of membranes in terms of biocompatibility and biodegradation after implantation, supporting future research in humans.

The variability in pore size in Groups 1 and 4 was due to the randomness of deposition and fiber diameter variation. The pores in the samples of Group 5 were more regular, as they followed the invariability of the fiber diameter, despite the random arrangement during their manufacture. It is believed that the variation in porosity favors cell adaptation and growth on the surface of the samples [35]. These characteristics corroborate the resorption pattern observed in the histological analysis.

Compared to the clinically available membrane (G5), the novel membranes presented good biocompatibility. G1 and G4 showed membranes surrounded by loosely arranged connective tissue in all experimental groups. From the first week, these membranes allowed the permeation of mesenchymal cells, populating inflammatory cells, and showed intense phagocytic activity, although they maintained their arrangement. In contrast, the membranes of G2 and G3 presented rectilinear and homogeneous aspects without adherence to the adjacent connective tissue. Additionally, at the periphery of the membrane, it was possible to observe mononuclear inflammatory cells and some giant cells “trying” to invade the membrane surface. This reaction corroborated the degradation mechanism of PTMC, which is cell-mediated enzymatic surface erosion [20,22] characterized by a mild foreign-body reaction.

With the objective of improving the mechanical and biological properties of PLGA, a series of associations have been reported in the literature, including PLGA + HA and -TCP [36], PLGA + hydroxyapatite hybrid nanofibrous scaffolds [37], and PLGA + collagen/chitosan (COL/CHI) [38]. Such associations can improve structural integrity and flexibility, act by releasing calcium, increase bone formation, and facilitate cell adhesion and spread.

In our study, PLGA was associated with PTMC. This combination, reported in the literature, showed promising results as biodegradable scaffolds based on polymeric materials [19,39]. PTMC is a polymer used in biomedical applications. It presents good properties of biocompatibility and flexibility. Additionally, it possesses good surface erosion properties, does not produce acid compounds after degradation, and shows slower degradation characteristics compared to PLGA [39]. Another exciting factor that points to PTMC as a good alternative for tissue engineering is its elastic characteristic at body temperature. Nevertheless, the inferior mechanical properties of PTMC have commonly hampered its widespread application. In this context, the PLGA and PTMC properties may be a good choice for the association of polymers [19,40].

A previous study conducted by de Santana et al. [41] evaluated the soft and hard tissue responses to the topographic characteristics of an absorbable synthetic polylactide membrane. The authors observed that the surface topographies promoted differential soft tissue responses, even on the same membranes. While the rough surface of the barrier contained significantly more giant cells, the smooth surface exhibited significantly more inflammatory cells. Similar findings were observed in the present study. Membranes of G1 and G4, manufactured by random fiber deposition and showing an irregular surface, presented a significant increase in giant cells compared to the control (*p* < 0.003). In contrast, Groups 2 and 3, manufactured by solvent casting with an impermeable pattern, presented a homogeneous and flat surface, with a greater population of lymphocytes than other groups (*p* < 0.05).

Therefore, we observed that the different presentations of membranes were biocompatible in the studied model. G2 and G3, which presented an impermeable character, maintained the tissue barrier pattern, preventing cell flow. This characteristic is related to the manufacturing method of solvent casting. Furthermore, there were no biological differences between G2 (100 µm) and G3 (70 µm) membranes. In contrast, the membranes of the G1 and G4 groups allowed cellular traffic due to their permeable character. In the membranes of G1, as they were narrower (100 µm), it was possible to observe greater and earlier cell flow and peripheral biodegradation than in G4. Group 4, given its greater thickness, delayed the process of cellular invasion. Both membranes had maintained their framework 12 weeks after implantation.

The investigation of cellular interactions performed in this study to investigate the compatibility of membranes is the initial step for evaluating the biological response in vivo. Additional research, including experimental bone tissue models, is suggested to explore key features before transferring the current findings to clinical practice.

## 5. Conclusions

The membranes are considered biocompatible. G5 showed a higher degree of biosorption, followed by G1 and G4. G2 and G3 are considered non-absorbable at the evaluated time points.

## Figures and Tables

**Figure 1 membranes-12-00711-f001:**
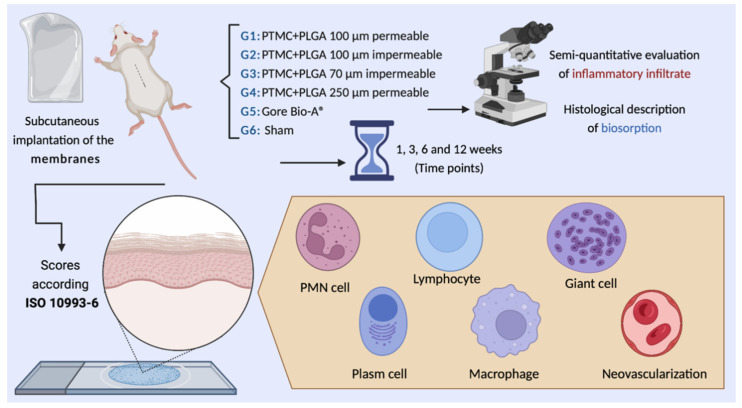
Graphical abstract of the experimental design. This figure was created with Biorender.com.

**Figure 2 membranes-12-00711-f002:**
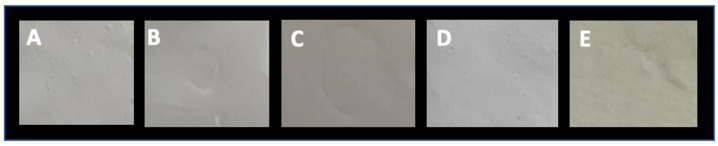
Macroscopic images of the membranes used in the study according to the experimental groups. (**A**): Group 1; (**B**): Group 2; (**C**): Group 3; (**D**): Group 4; (**E**): Group 5.

**Figure 3 membranes-12-00711-f003:**
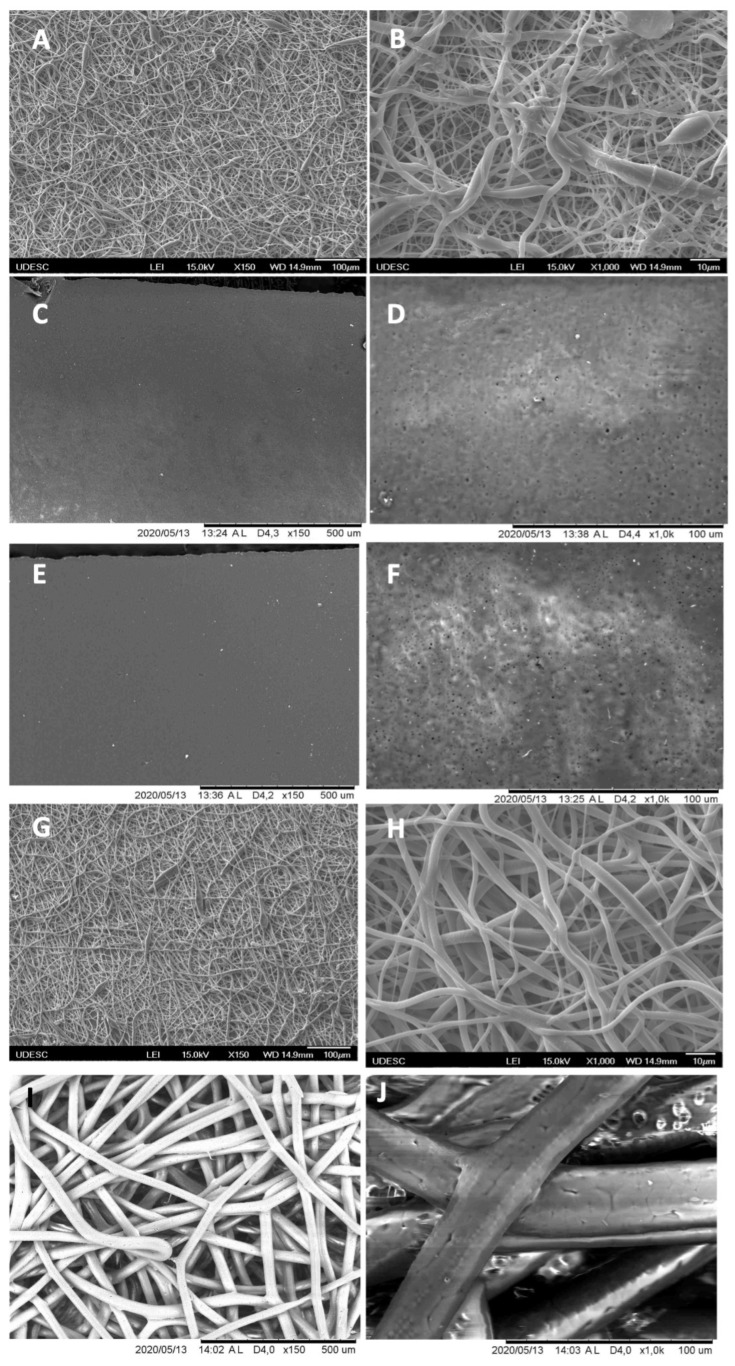
Scanning electron microscopy (SEM) micrographs according to the experimental groups. (**A**,**B**): Group 1; (**C**,**D**): Group 2; (**E**,**F**): Group 3; (**G**,**H**): Group 4; (**I**,**J**): Group 5. (**A**,**C**,**E**,**G**,**I**): 150× magnification; (**B**,**D**,**F**,**H**,**J**): 1000× magnification.

**Figure 4 membranes-12-00711-f004:**
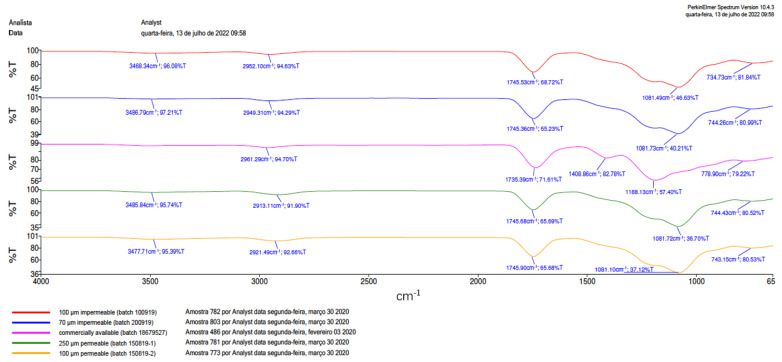
Fourier transform infrared (FTIR) spectrums. The FTIR spectrum showed vibrational modes typical of PTMC + PLGA (Groups 1–4) and PLG + PTMC (Group 5).

**Figure 5 membranes-12-00711-f005:**
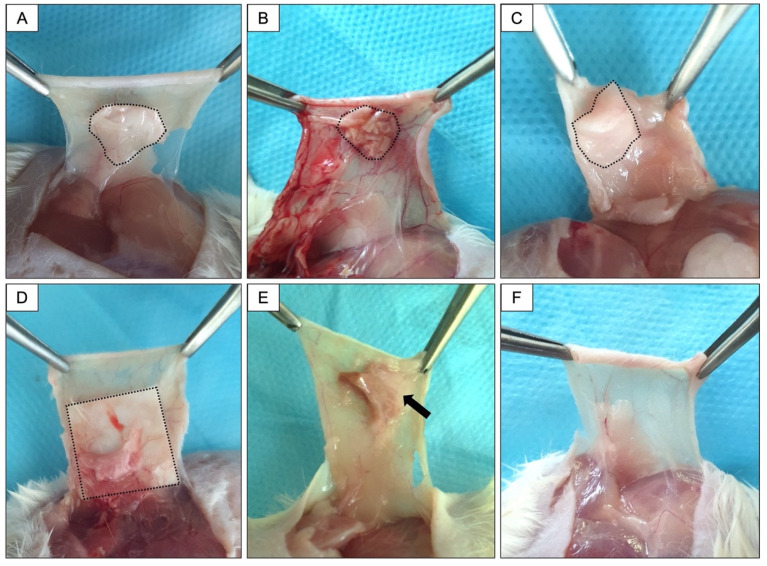
Macroscopic aspect of the tissue response to different membranes after 12 weeks of implantation. (**A**): Group 1; (**B**): Group 2; (**C**): Group 3; (**D**): Group 4; (**E**): Group 5; (**F**): Sham group. The dotted area corresponds to the membrane region. The black arrow indicates the granulation reaction present in Group 5, where the membrane was not visualized.

**Figure 6 membranes-12-00711-f006:**
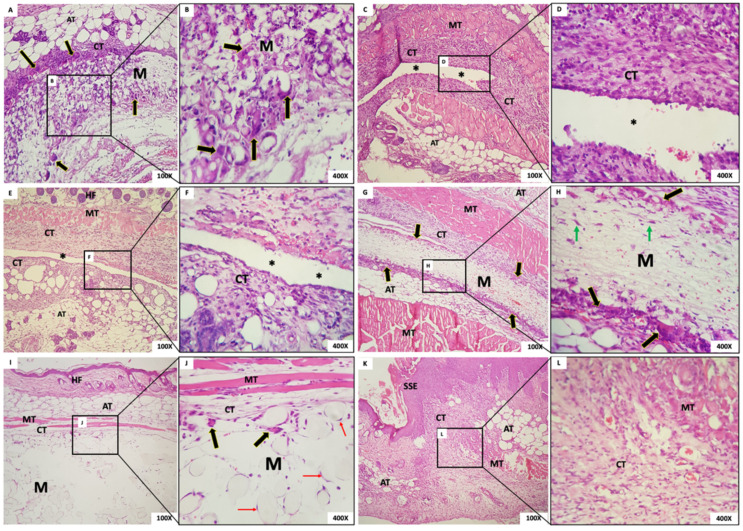
(**A**,**B**) Group 1; (**C**,**D**) Group 2; (**E**,**F**) Group 3; (**G**,**H**) Group 4; (**I**,**J**) Group 5 (**K**,**L**). Group sham. Membrane (M); connective tissue (CT); multinucleated giant cells (black arrows); adipose tissue (AT); empty space (*); muscle tissue (TM); hair follicles (HF); mesenchymal cells (green arrows); mesenchymal cells surrounding the membrane spheres (red arrows); hyperplastic stratified squamous epithelium (SSE). Scale bar: (**A**,**C**,**E**,**G**,**I**,**K**): 100 µm and (**B**,**D**,**F**,**H**,**J**,**L**): 400 µm. Staining: Hematoxylin and Eosin.

**Figure 7 membranes-12-00711-f007:**
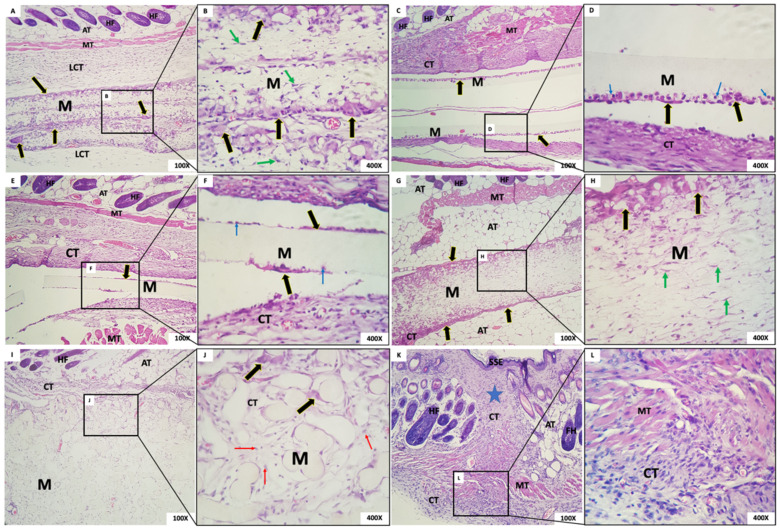
(**A**,**B**) Group 1; (**C**,**D**) Group 2; (**E**,**F**) Group 3; (**G**,**H**) Group 4; (**I**,**J**) Group 5; (**K**,**L**) Sham group. Membrane (M); loose connective tissue (LCT); multinucleated giant cells (black arrows); adipose tissue (AT); hair follicles (HF); muscle tissue (TM); mesenchymal cells (green arrows); connective tissue (CT); mononuclear cells inside the membrane (blue arrows); mesenchymal cells surrounding the membrane spheres (red arrows); orthokeratinized stratified squamous epithelium (SSE); area of incision (Star). Scale bar: (**A**,**C**,**E**,**G**,**I**,**K**): 100 µm and (**B**,**D**,**F**,**H**,**J**,**L**): 400 µm. Staining: Hematoxylin and Eosin.

**Figure 8 membranes-12-00711-f008:**
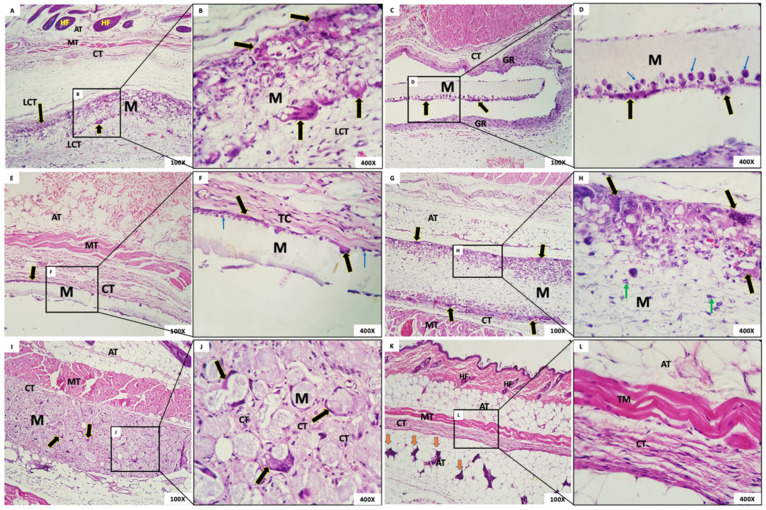
(**A**,**B**) Group 1; (**C**,**D**) Group 2; (**E**,**F**) Group 3; (**G**,**H**) Group 4; (**I**,**J**) Group 5; (**K**,**L**) Sham group. Membrane (M); loose connective tissue (LCT); multinucleated giant cells (black arrows); adipose tissue (AT); hair follicles (HF); muscle tissue (TM); granulation reaction (GR); mononuclear cells inside the membrane (blue arrows); mesenchymal cells (green arrows); connective tissue (CT); Glandular tissue (orange arrows). Scale bar: (**A**,**C**,**E**,**G**,**I**,**K**): 100 µm and (**B**,**D**,**F**,**H**,**J**,**L**): 400 µm. Staining: Hematoxylin and Eosin.

**Figure 9 membranes-12-00711-f009:**
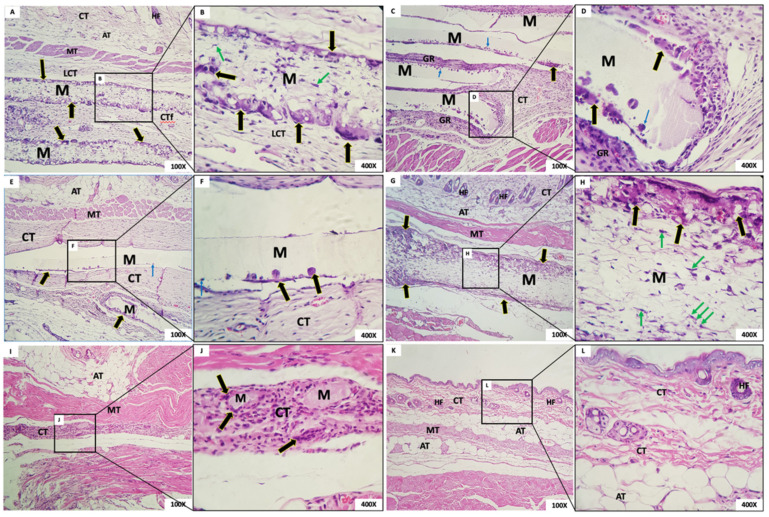
(**A**,**B**) Group 1; (**C**,**D**) Group 2; (**E**,**F**) Group 3; (**G**,**H**) Group 4; (**I**,**J**) Group 5; (**K**,**L**) Sham group. Membrane (M); loose connective tissue (LCT); multinucleated giant cells (black arrows); connective tissue (CT); adipose tissue (AT); hair follicles (HF); muscle tissue (TM); mesenchymal cells (green arrows); granulation reaction (GR); mononuclear cells inside the membrane (blue arrows). Scale bar: (**A**,**C**,**E**,**G**,**I**,**K**): 100 µm and (**B**,**D**,**F**,**H**,**J**,**L**): 400 µm. Staining: Hematoxylin and Eosin.

**Figure 10 membranes-12-00711-f010:**
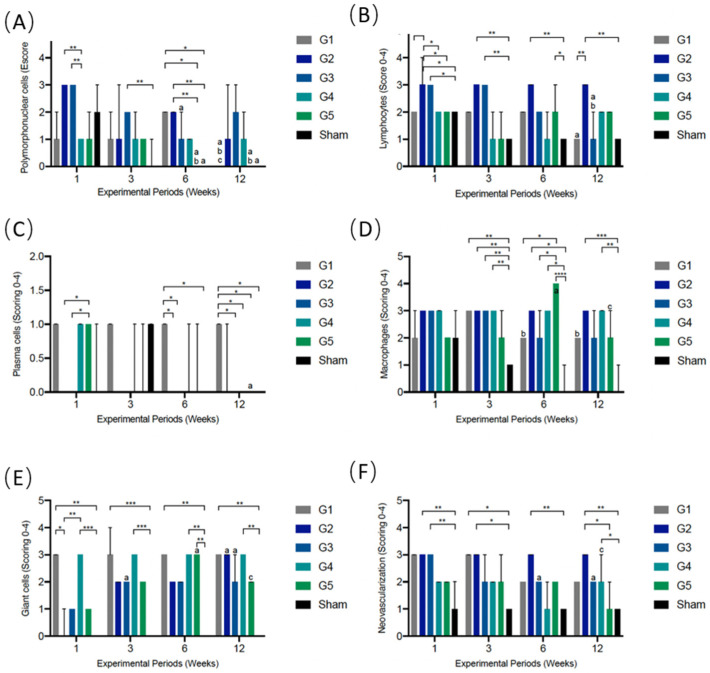
Inflammatory cell response and overall tissue reaction (**A**–**F**) after experimental periods of 1, 3, 6, and 12 weeks. The values are presented as median ± confidence interval. The horizontal bars represent significant differences between different groups at the same experimental time point (Kruskal–Wallis and Dunn’s post hoc tests; *p* < 0.05). The letters represent significant differences between time points with the same treatment. (a) The significant difference compared to 1 week; (b) significant difference compared to 3 weeks (c) significant difference compared to 6 weeks (Kruskal–Wallis and Dunn’s post hoc tests; *p* < 0.05). (* *p* = 0.01–0.04; ** *p* = 0.001–0.009; *** *p* = 0.0001–0.0006; **** *p* < 0.0001).

## Data Availability

Not applicable.

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
