# Peer review of "In Vivo Evaluation of Permeable and Impermeable Membranes for Guided Bone Regeneration"

_membranes, 2022, doi:10.3390/membranes12070711_

Round 1
Reviewer 1 Report
The manuscript presented the evaluation of permeable and impermeable membranes for GBR using new synthetic materials. several questions need to be clarified / explained.
(1) The authors state that " G2 had a major inflammatory reaction", however, the membrane is considered biocompatible. Could the authors explain this result more clearly? How was the biocompatibility assessed? In accordance to ISO 10993-5?
(2) Two different fabrication techniques are mentioned to prepare permeable and impermeable membranes. Could the authors provide more details on how these two techniques work in the Materials and Methods section.
(3) Different pore sizes are prepared. How were the sizes chosen? Is there any explanation as to why 70, 100, 250 µm were considered in the study.
(4) Can the authors show the ratings of the 5 tested membranes for their intended use? How will the permeability or impermeability affect the GBR? Is it possible to explain the different results of incorporating the underlying tissue based on the presented characteristics?
Author Response
Dear Professor Spas D. Kolev
Editor-in-Chief of Membranes
Ref.: Manuscript: membranes-1778501
Title: In Vivo Evaluation of permeable and impermeable membranes for Guided Bone Regeneration
Thank you for your attention to our manuscript “In Vivo Evaluation of permeable and impermeable membranes for Guided Bone Regeneration.” Indeed, we appreciated the comments and criticism of the reviewer and the opportunity you gave us to submit a new revision of our reworked manuscript.
The authors would like to acknowledge the effective and unbiased review of the manuscript, believing that introducing the suggested alterations produced a manuscript with better editorial and scientific quality.
Please find below the point-by-point answers to reviewers’ comments:
REVIEW #1
The manuscript evaluated permeable and impermeable membranes for GBR using new synthetic materials. several questions need to be clarified / explained.
(1) The authors state that "G2 had a major inflammatory reaction", however, the membrane is considered biocompatible. Could the authors explain this result more clearly? How was the biocompatibility assessed? In accordance to ISO 10993-5?
Answer: Our study aimed to evaluate the biocompatibility and biosorption of different membranes and compare them to each other. For this evaluation, we conducted a semiquantitative Histological Analysis of the local biological effect of implanted membranes: ISO 10993-6: 2016/Part 6/Annex E. The details were included in 2.7.2. section and is described at the end of this answer.
In a comparative evaluation, membrane two was considered to have greater inflammation than the other membranes as it had a higher inflammatory response score in different scenarios. However, in an individual assessment, the inflammatory reaction was not considered exuberant since the group reached the maximum score of cell recruitment or signs of necrosis and/or fatty infiltrate in nonexperimental periods.
As described in the results sections 3.4.1 and 3.4.2, the G2 recruited more PMNs cells than group 4 (p=0.004) after one week and showed greater cell recruitment than groups G5 and sham, After six weeks. Also, in general, the G2 presented a greater lymphocyte cell volume than other groups.
Regarding the PMNs evaluation, it’s important to point out that the maximum score (4 – Packed) was not reached in any experimental group/period. In addition, after one week, groups 2 and 3 reached a score of 3, without differences when compared to the sham group. In other words, against the different membranes, the tissue responded mildly, without acute inflammation or infection; thus, the membranes were considered biocompatible.
Like neutrophils, the lymphocytes did not show a maximum score in any group/experimental period. The high levels of mononuclear cell recruitment were observed in G2, which remained as score heavy infiltrate during all experimental periods (p<0.01). This reaction can be explained by the greater cell signaling in membranes of difficult incorporation/phagocytosis, consequently a more prominent and persistent granulation reaction. The authors have included two paragraphs in Material and Methods (2.7.2) explaining how the biocompatibility was assessed.
(2) Two different fabrication techniques are mentioned to prepare permeable and impermeable membranes. Could the authors provide more details on how these two techniques work in the Materials and Methods section.”
Answer: Permeable membranes: polymeric membranes of randomly deposited fibers forming a three-dimensional mesh. In manufacturing, the polymer is dissolved in an organic solvent and subjected to a high (negative) electric field. When the applied electric field overcomes the polymer solution's surface tension and viscous forces, a charged jet of the solution is ejected toward the opposite pole (collector), also electronically charged, but with an opposite charge (positive), thus forming the fibers. During the path between the polymer solution and the fiber collection surface, the jet presents instability regions, evaporating the solvent and depositing solid fibers on the collector (electrospinning).1 Then, the resulting membrane is removed from the collector and cut according to the need and indication of use.
Impermeable membranes are polymeric barriers manufactured by preparing a polymer solution in an organic solvent.2,3 After total homogenization of the solution, it is placed in polyacetal molds with the internal thickness required for the barrier. The molds are vacuumed for 3 hours for complete evaporation of the solvent and drying of the barriers. Then, the drying process is completed, and the barriers are removed from the molds and ready for cutting, according to the need and indication of use.
(3) Is there any explanation as to why 70, 100, 250 µm were considered in the study.
Answer: The criteria for defining the thicknesses of the samples were their handling and behavior during the surgical procedures, which were tested in a previous study. It is essential that the surgeon can successfully cover the bed required for tissue reconstruction, and stability (immobility) is necessary for the healing to occur properly. Therefore, the membrane must allow adaptation to the bed (flexibility) and sufficient resistance to be stabilized by sutures, screws, or surgical pins. The membranes produced with thicknesses of 70, 100, and 250 µm have also demonstrated in previous simulation tests in vitro that they meet these characteristics.
(4) Can the authors show the ratings of the 5 tested membranes for their intended use? How will the permeability or impermeability affect the GBR? Is it possible to explain the different results of incorporating the underlying tissue based on the presented characteristics?
Answer: Nowadays, impermeable non-resorbable membranes (e.g., high-density polytetrafluorethylene - d-PTFE) are available, indicated to remain exposed to the oral environment, if necessary, as in cases of preservation of alveolar volume after exodontia. The objective of producing and comparing resorbable permeable and impermeable membranes was to evaluate if there is a difference in the capacity of tissue regeneration, in this case of bone tissue, for future developments, as an indication of exposure to a permeable and resorbable membrane. The samples used in this study have an indication for Guided Tissue Regeneration (GTR), not only Guided Bone Regeneration (GBR).
Reviewer 2 Report
OPINION: I am of positive opinion with the following recommendations.
In the view of this reviewer, despite the introduction being very well written, the background on GBR is very unnecessary, leaving the background too long, in addition there are several literature reviews that address it in detail. The first 2 paragraphs; they should be removed!! Kindly review the whole INTRO to become more concrete & merge the related paragraphs together.
1-) In the introduction, authors should focus on the key elements to support the working hypothesis. I believe that the presentation to the readers about the difference between the types of membranes would be the starting point and the authors should focus more on the chemical part of the material with the possible characteristics and interaction with the biological.
2-) Material and method section is inadequate, it needs to separate and include the items used, in addition to designating the information of each input, reagent used. However, if adding in the text, the information must appear in parentheses.
Specify the origin (city, state, country and brand, batch) of all pharmaceutical ingredients and solvents used in the preparations, as well as the equipment used (model, city, state, country and brand, serial number.
3-) Add ethics committee approval date
4-) The authors did not add the methodology for preparing the membranes. Item that must be mandatory at work. How did the researchers manage to reproduce the material without having the details?
5-) Figure 2 could have been prepared so that the membranes were all aligned
6-) The FITR methodology lacks information on the amount of sample used, if it was mixed with KBr ou was used FTIR-ATR.
7-) How many study groups? how many animla/group? no appropriate sham control was conducted containing empty membranes.
8-) 2.7.1 standardize the italic font
9-) SEM – What size of the pore? What size of the fibers? What the roughness? It is very easy to find these values with any software, the authors should add to improve the discussion of the results.
10-) Figure 4 of FTIR needs to be better treated, as the authors plotted, it is not possible to distinguish the bands.Please review the FITR articles from the literature and see how to separate the spectra.Furthermore, the authors need to highlight the bands and present the chemical groups. The discussion is very poor.
11-) I apologize, but I am not able to analyze the results in vivo.
Author Response
Dear Professor Spas D. Kolev
Editor-in-Chief of Membranes
Ref.: Manuscript: membranes-1778501
Title: In Vivo Evaluation of permeable and impermeable membranes for Guided Bone Regeneration
Thank you for your attention to our manuscript “In Vivo Evaluation of permeable and impermeable membranes for Guided Bone Regeneration.” Indeed, we appreciated the comments and criticism of the reviewer and the opportunity you gave us to submit a new revision of our reworked manuscript.
The authors would like to acknowledge the effective and unbiased review of the manuscript, believing that introducing the suggested alterations produced a manuscript with better editorial and scientific quality.
Please find below the point-by-point answers to reviewers’ comments:
REVIEW #2
In the view of this reviewer, despite the introduction being very well written, the background on GBR is very unnecessary, leaving the background too long, in addition there are several literature reviews that address it in detail. The first 2 paragraphs; they should be removed!! Kindly review the whole INTRO to become more concrete & merge the related paragraphs together.
1-) In the introduction, authors should focus on the key elements to support the working hypothesis. I believe that the presentation to the readers about the difference between the types of membranes would be the starting point and the authors should focus more on the chemical part of the material with the possible characteristics and interaction with the biological.
Answer: We appreciate the reviewer's observation; the first two paragraphs were removed. Also, the third paragraph, which linked to the first two, was adjusted for better understanding.
The difference between the types of membranes and the chemical part of the material with the possible characteristics and interaction with the biological was included in the introduction section.
Both are composed of biodegradable polymers in the form of PLGA (poly-D, L-lactic/glycolic acid) and Poly (L-lactate-co-Trimethylene Carbonate) (PTMC). These polymers are biodegradable and have been used for years in the medical industry; they degrade in vivo and are then metabolized for energy, resulting in H2O and CO2.
The difference between the two (permeable and impermeable) is in the physical structure since the chemical composition is the same.
The permeable membrane has a porous, nanostructured, three-dimensional surface, favoring cell adhesion but preventing cells from crossing it. The pores allow the passage of nutrients through the membrane to the area under regeneration. Therefore, it is important that the membrane is covered by the tissue surrounding the area to be regenerated so that no external interferences may cross the membrane, harm the regenerative process, or even accelerate its degradation process.
On the other hand, the impermeable membrane prevents both the passage of nutrients and cell migration. It is a contention for the tissue, preventing external agents from interfering in the regenerative process. One suggestion is that it is indicated in cases where there is not enough surrounding tissue to cover the area that is intended to regenerate, such as non-absorbable membranes of high-density polytetrafluoroethylene (d-PTFE). The possibility of exposure to the environment (e.g., oral environment) should be investigated and ratified in future trials.
2-) Material and method section is inadequate, it needs to separate and include the items used, in addition to designating the information of each input, reagent used. However, if adding in the text, the information must appear in parentheses.
Specify the origin (city, state, country and brand, batch) of all pharmaceutical ingredients and solvents used in the preparations, as well as the equipment used (model, city, state, country and brand, serial number.
Answer: Thanks for the comment. The authors have added the designating the information of each material used in parentheses.
3-) Add ethics committee approval date
Answer: The authors have included the ethics committee approval date.
4-) The authors did not add the methodology for preparing the membranes. Item that must be mandatory at work. How did the researchers manage to reproduce the material without having the details?
Answer: The authors have added the text's methodology for preparing the membranes. Permeable membranes: polymeric membranes of randomly deposited fibers forming a three-dimensional mesh. In manufacturing, the polymer is dissolved in an organic solvent and subjected to a high (negative) electric field. When the applied electric field overcomes the surface tension and viscous forces of the polymer solution, a charged jet of the solution is ejected toward the opposite pole (collector), also electronically charged, but with an opposite charge (positive), thus forming the fibers. During the path between the polymer solution and the fiber collection surface, the jet presents instability regions, evaporating the solvent and depositing solid fibers on the collector (electrospinning). Then, the resulting membrane is removed from the collector and cut according to the need and indication of use. Impermeable membranes are polymeric barriers manufactured by preparing a polymer solution in an organic solvent. After total homogenization of the solution, it is placed in polyacetal molds with the internal thickness required for the barrier. The molds are vacuumed for 3 hours for complete evaporation of the solvent and drying of the barriers. Then, the drying process is completed, and the barriers are removed from the molds and ready for cutting, according to the need and indication of use.
5-) Figure 2 could have been prepared so that the membranes were all aligned
Answer: The authors have aligned the membranes in a new Figure 2.
6-) The FITR methodology lacks information on the amount of sample used, if it was mixed with KBr or was used FTIR-ATR.
Answer: One membrane per group (1x1 cm) was used for the FTIR evaluation. The membranes were arranged on the infrared crystal and compressed at 100 N to overcome the radiation and obtain the data for forming the FTIR bands. No KBr was used in the preparation of the samples. This information was included in the FTIR methodology.
7-) How many study groups? how many animla/group? no appropriate sham control was conducted containing empty membranes.
Answer: In this study, we used Sixty Balb-C mice randomly divided into six experimental groups and subdivided into one, three, six, and 12 weeks (n=5 groups/period). The experimental groups were divided into four different experimental versions of Poly (L-lactate-co-Trimethylene Carbonate) (PTMC) + Poly (L-lactate-co-glycolate) (PLGA) membranes, a commercially available membrane, and the Sham group (only surgical procedure, without implantation) as follows:
Group 1: 100 µm thickness permeable (batch 150819-2);
Group 2: 100 µm thickness impermeable (batch 100919);
Group 3: 70 µm thickness impermeable (batch 200919);
Group 4: 250 µm thickness permeable (batch 150819-1);
Group 5: commercially available - Gore Bio-Aâ (batch 18679527);
Group 6: Sham (without implantation).
This information was added in the Abstract/Graphic Abstract, Introduction, and Animal Characterization sections.
8-) 2.7.1 standardize the italic font
Answer: The authors have standardized the italic font during the text.
9-) SEM – What size of the pore? What size of the fibers? What the roughness? It is very easy to find these values with any software, the authors should add to improve the discussion of the results.
Answer: Unfortunately, we have not received a response from the laboratory that performed the analysis to measure the pore sizes. According to the information on the images, we can state that the pores have a size of ≤ 0.04µm. If this reviewer considers the measurement in the software extremely important, we will need more time for further contact.
10-) Figure 4 of FTIR needs to be better treated, as the authors plotted, it is not possible to distinguish the bands. Please review the FITR articles from the literature and see how to separate the spectra. Furthermore, the authors need to highlight the bands and present the chemical groups. The discussion is very poor.
Answer: The bands of the membranes in the test groups confirmed their same composition. Comparing the bands of the control and tested groups, two different absorption peaks were observed: 3468-3486 presented in the membranes of the test groups and absent in the membrane of the control group, which represent the functional group O-H (alcohol), and 1408 present in the membrane of the control group and absent in the membranes of the test groups, which represents the functional group O-H (carboxylic acid). These differences are expected due to the composition of the membranes of the test and control groups not being identical. However, both functional groups will be degraded in the same way, with the organism's H2O molecule as a byproduct. Figure 4 was redone for better visualization of each sample.
Round 2
Reviewer 2 Report
The authors answered the initial questions. The work is adequate and contributes to science. I accept in this present form.